# Viewing Aggregation-Induced Emission of Metal Nanoclusters from Design Strategies to Applications

**DOI:** 10.3390/nano13030470

**Published:** 2023-01-24

**Authors:** Tingting Li, Haifeng Zhu, Zhennan Wu

**Affiliations:** 1School of Materials Science and Engineering, Jilin Jianzhu University, Changchun 130018, China; 2State Key Laboratory of Integrated Optoelectronics, College of Electronic Science and Engineering, Jilin University, Changchun 130021, China

**Keywords:** aggregation-induced emission, metal nanoclusters, luminescent nanoclusters

## Abstract

Aggregation-induced emission (AIE)-type metal nanoclusters (NCs) represent an innovative type of luminescent metal NCs whose aggregates exhibit superior performance over that of individuals, attracting wide attention over the past decade. Here, we give a concise overview of the progress made in this area, from design strategies to applications. The representative design strategies, including solvent-induction, cation-induction, crystallization-induction, pH-induction, ligand inheritance, surface constraint, and minerals- and MOF-confinement, are first discussed. We then present the typical practical applications of AIE-type metal NCs in the various sectors of bioimaging, biological diagnosis and therapy (e.g., antibacterial agents, cancer radiotherapy), light-emitting diodes (LEDs), detection assays, and circularly polarized luminescence (CPL). To this end, we present our viewpoints on the promises and challenges of AIE-type metal NCs, which may shed light on the design of highly luminescent metal NCs, stimulating new vitality and serving as a continuous boom for the metal NC community in the future.

## 1. Introduction

Metal nanoclusters (NCs) typically possess sub-2 nm metal cores consisting of a few to a hundred metal atoms that are protected by an organic ligand monolayer [1,2,3]. They are a rising star in the nanoparticle community due to their significance in both basic and applied research (e.g., as the missing link between atoms and nanocrystals, for the utilization of their intriguing molecular-like properties, including discrete electronic transitions, and for quantized charging) [4,5,6,7]. Among various properties, luminescence is the most attractive property of metal NCs. Because of their facile preparation, ultrafine size, low toxicity, high renal clearance, and excellent photostability, luminescent metal NCs have recently emerged as a novel class of chromophores, holding great promise for practical applications in lighting, imaging, sensing, and so on [8,9,10,11,12,13].

However, highly luminescent metal NCs are curtailed by the comparatively limited knowledge of their fundamental aspects. In particular, the complexity, diversity, and mutability in terms of their total structures preclude an in-depth understanding of their emission origin. Because of their unclear size/composition effects and the puzzling underlying mechanism of their luminescence, there has been limited success in attaining the desired improvement and tailoring the luminescence property of metal NCs, thus greatly limiting their boom in practical applications [14,15,16]. In this scenario, with continuous efforts in the community, Xie et al. first presented the aggregation-induced emission (AIE) of metal nanoclusters in 2012, which allows for an extraordinary strength in their emission intensity and stability [17]. In detail, the authors designed a new family of ultrabright Au(0)@Au(I)-SR (SR: deprotonated thiol ligands) core–shell NCs by preserving a high content of Au(I)-SR complexes in the protective shell. In this way, the intra-/intermolecular vibration and rotation of surface motifs can be effectively suppressed, giving rise to a minimization of non-radiative decay and hence the improved luminescence performance of metal NCs. 

Thereafter, AIE-type metal NCs have been a recognized concept and a well-developed topic in both the fundamental and practical sectors of luminescent metal NCs (Figure 1). In the scheme of AIE-type metal NCs, over the past decade, several significant design strategies have been identified at and beyond the single-cluster level [18,19,20]. Their practical applications have been well proven in broad fields and cross-disciplines. Therefore, we deploy this review by viewing AIE-type metal NCs from their design strategies to their practical applications (Figure 2). Together with our overviews on the promises and challenges of AIE-type metal NCs, we try to offer a brief and in-depth discussion on AIE-type metal NCs in order to increase the acceptance of AIE-type metal NCs in various related communities in chemistry and materials science.

## 2. Design Strategies of AIE-Type Metal NCs

AIE is an extraordinary luminescence concept, which was first proposed by Tang in 2001 to illuminate an interesting photophysical phenomenon: the polymeric state is brighter than the dispersed state [21]. In particular, restricted intramolecular motion (RIM) has been proved to be the general mechanism of AIE [22]. Normally, many AIE molecules contain molecular rotors and/or vibrators. In a solution state, these rotors and vibrators can rotate and vibrate flexibly to consume the excited state energy, resulting in AIE molecules being nonemissive or weakly emissive. Meanwhile, in an aggregate state, intermolecular interactions restrict the rotation and vibration, causing radiative decay to dominate the energy dissipation of the excited state [23,24,25].

Since Xie et al. introduced the concept of AIE into metal NCs in 2012 [17,18,19,26], the family of AIE-type metal NCs has grown over the past decade: from a single-cluster level to the beyond-single-cluster level, from small molecules to macromolecules, from the solution state to the crystalline state, from condensed oligomeric complexes to scaffold confinement-induced materials, etc. Accordingly, in this section, we rationally categorize and discuss AIE-type metal NCs by focusing on their construction strategies.

### 2.1. Solvent-Induced AIE

Solvent-induced AIE is a representative strategy to trigger the aggregates from isolated metal NCs by generally altering the solvent polarity. Xie et al. successfully designed the first AIE-Au(0)@Au(I)-SR NCs based on this strategy by introducing ethanol (a weakly polar solvent) into water [17]. The formation of aggregates can be attributed to two reasons: (i) the addition of the ethanol destabilizes the Au(I)−thiolate compound, resulting in charge neutralization; (ii) the amplification of the Au(I)···Au(I) interaction to provide momentum for aggregation. The luminescence properties of aggregates strongly depend on the aggregation degree, which can be expressed by the formula *f*_e_ = vol_ethanol_/vol_ethanol+water_. The solution exhibited non-emission before *f*_e_ > 75%; when *f*_e_ increases from 75% to 95%, the aggregates began to form, emitting weak red radiation; finally, the solution turned clarified and emitted very strong yellow luminescence at *f*_e_ > 95%, as shown in Figure 3a–c. Upon aggregate formation, owing to the multiple intra- and inter-complex interactions (Au(I)···Au(I) interactions and van der Waals forces), intramolecular rotations and vibrations are limited, therefore enhancing the luminescence. Moreover, the blueshift mechanism is interpreted as the increase in Au(I)···Au(I) distance, because the inter-complex aurophilic interactions are stronger than intra-complex interactions. 

Au(I)*_x_*(SR)*_x_*_+1_ (SR = glutathione), recently proposed by Xie et al., is another typical case to demonstrate the importance of solvents for the preparation of AIE-type metal NCs [26]. Serial AIE-type Au NCs were reported in this work, that is, the aggregates of Au_10_(SR)_10_, Au_15_(SR)_13_, Au_18_(SR)_14_, and [Au_25_(SR)_18_]^−^ (Figure 3d). Except for [Au_25_(SR)_18_]^−^ NCs, the quantum yield of Au_10_(SR)_10_, Au_15_(SR)_13_, and Au_18_(SR)_14_ was heightened at 11.4%, 4.9%, and 3.7%, respectively, as *f*_e_ increased from 75% to 95%. The strong emission can be attributed to the restricted molecular motion due to the adequate interactions between Au(I)-thiolate motifs. As for emission energy, Au_10_(SR)_10_, Au_15_(SR)_13_, and Au_18_(SR)_14_ blue-shifted by 15, 9, and 6 nm, respectively, and [Au_25_(SR)_18_]^−^ red-shifted by 167 nm. The average Au^I^-Au^I^ distance is affected by the competition with intra- and inter-motif aurophilic interactions, which ultimately influences the emission peak position. A similar phenomenon was observed in Cu NCs by the Patra group, where the solvent-induced aggregation strategy was adopted to tune the emission peak and enhance the emission intensity of Cu_34−32_(SG)_16−13_ (Figure 3e,f) [27]. When *f_e_* = 90%, the emission intensity enhanced 36-fold, which was associated with a 28 nm blue-shifting (624 → 597 nm).

Sugiuchi et al. reported the AIE phenomenon on a diphosphine-ligated [Au_6_]^2+^ cluster by regulating the water contents (WC) [28]. The emission intensity decreased first and then increased by increasing the WC from 0% to 90%. Specifically, the aggregation at the water-rich conditions increased luminescence by 20 times (Figure 3g). The luminescent performance of the aggregates is highly dependent on the WC, which can be divided into the following three zones: (i) Zone A (WC < 40%), in which the cluster molecules separate off from each other and randomly orient themselves due to the weak attractive hydrophobic interactions (Figure 3h(i)). The concentration quenching would occur at this stage. (ii) Zone B (40% < WC < 70%) is the transition stage of zone A to zone C, in which the cluster molecules will ulteriorly agglomerate, resulting in severe concentration quenching. (iii) In zone C (70% < WC < 90%), the clustered molecules are tightly packed, thus suppressing the molecular motion and vibration to enhance luminescence (Figure 3h(ii)). Very recently, the Li group also discovered the AIE phenomenon in Cu(I)-based cyclic trinuclear complexes (Cu(I)-CTC) [29]. These NCs exhibit no photoluminescence in pure tetrahydrofuran until water is added. In particular, Cu(I)-CTC with a near-unity PLQY was obtained when vol_water_/vol_water+THF_ = 90%. As a result, the molecular forces including C−H···C/C−H···π interaction are the main reasons for the cluster aggregation and thus the non-radiative decay reduction.

### 2.2. Cation-Induced AIE

Like solvent-induced AIE, cation-induced AIE of metal NCs has also been well developed, due to the variations, rotations, and other movements of ligands and complexes that can be restrained by surface charge neutralization and crosslinking. In particular, silver-doped Au(0)@Au(I)-thiolate NCs are an important example of cation-induced AIE; their emission is greatly enhanced by four times [30]. The Ag-doped Au NCs’ lifetimes are 2.03, 5.16, and 52.2 μs, which is longer than pristine Au NCs, demonstrating that the additive silver can crosslink Au^I^-GSH motifs to facilitate the evolution of denser aggregates. A similar phenomenon was observed in the system of [Ag_x_Au_25-x_(PPh_3_)_10_(SC_2_H_4_Ph)_5_Cl_2_]^2+^ (200-fold quantum yield boosting) [31] and Au-doped Ag_29_(BDT)_12_ (TPP)_4_ (26-fold quantum yield boosting) NCs [6]. 

In addition to coinage metal ions, group IIB metal ions (Cd^2+^ and Zn^2+^) have also been used to promote the AIE process. Xie et al. utilized Cd^2+^ to induce the aggregation of Au(I)-thiolate complexes based on the electrostatic and coordination interactions between Cd^2+^ and GSH [17]. Kuppan et al. observed that non-luminescent 3-mercaptopropionic acid Au NCs can achieve strong yellow emission (10^6^-fold improvement) by the incorporation of Zn^2+^ [32]. As depicted in Figure 4a, the self-assembly of Au NCs was efficiently induced by Zn^2+^ through two main ways: (i) the strong complexation of Zn^2+^ with negatively charged carboxyl (i.e., COO^−^) groups by the charge screening effect, enhancing the rigidity of the Au(I) thiolate shell; (ii) Zn^2+^ crosslinking with surrounding Au NCs to form tight aggregates upon the force of van der Waals interactions. These interactions hinder the vibrations and rotations of ligands, significantly decreasing the non-radiative relaxation. Another example of a Zn^2+^ cation-induced AIE is based on Au_4_(MHA)_4_ (MHA = 6-mercaptohexanoic acid) NCs [16]. The aggregated Au_4_(MHA)_4_ exhibits ultrabright greenish-blue emission centered at 485 nm, with a narrow FWHM of 25 nm and a high QY of 90%. Similarly, the assembled mechanism is attributed to the interactions between the COO^−^ in the MHA ligand and Zn^2+^, which rigidify the surrounding environment of Au_4_(MHA)_4_ NCs. 

Recently, researchers have focused their attention on the rare earth ions. Luminogen ATT-AuNCs (ATT: 6-aza-2-thiothymine) are discussed here as a representation [33]. It was discovered that the luminescence properties of ATT-Au NCs significantly depend on the pH. Decreasing the pH results in the red-shift of emission spectra (from 520 to 535 nm) and significant PL quenching (Figure 4b). Sc^3+^ exhibits Lewis acidity and can stabilize the ICT state, thus being used to induce aggregation of ATT-Au NCs. The addition of Sc^3+^ not only makes the spectrum red-shift, but also promotes the PLQY enhancement (from 0.2 to 5.5%). As a result, the role of Sc^3+^ mainly includes three aspects: enhancing the ICT state; triggering the interaction between Sc^3+^ and the surface ligand to form the RIM process; constructing a donor–bridge–acceptor (DBA) structure by linking ATT-AuNCs and minocycline (Figure 4c,d).

### 2.3. Crystallization-Induced AIE

Crystallization-induced emission (CIE) may be treated as a derivative of AIE. In general, the active intramolecular motion (vibrations and rotations) in the solution state dissipates the energy of the excited state through non-radiative relaxation channels, leading to luminescence quenching. However, the intramolecular motion is restricted in the solid state, which minimizes energy consumption [34]. Crystallization can achieve a transition from liquid to solid, that is, by physical constraints and intermolecular interactions to lock the molecular conformations and hold them together. Zhu et al. presented the first CIE metal clusters in 2017. The bimetallic NCs were protected by (diphenylphosphino)methane (DPPM) and 2, 5-dimethylbenzenethiol (SR), which provided a guiding strategy to build up the AIE-type metal NCs [35]. The solution of Au_4_Ag_13_(DPPM)_3_(SR)_9_ is almost non-emissive (QY: ∼4.1 × 10^−5^), whether in CH_2_Cl_2_ or MeOH, but its amorphization and crystallization are fluorescent. The QY value of crystallization and amorphization is 653-fold and 278-fold higher than that of the solution state, respectively. In the solution state, the weak interactions and random collisions of the solvent dissipate the exciton energy and reduce luminescence. In the crystalline state, however, the inherent tri-blade fan configuration and multiple C-H⋯π interactions effectively restrict the molecular vibrations and rotations. CIE sliver-based NCs, [Ag_22_(dppe)_4_(SR)_12_Cl_4_]^2+^, were discovered in subsequent years by the Pradeep group. There exist multiple C−H···π and π···π interactions between phenyl groups in these NCs’ crystals, enhancing their luminescence in the crystalline state by 12 times compared to the amorphous state [36]. Two examples of CIE copper-based NCs, Cu_2_(C_3_H_3_N)_6_I_2_ (1) and Cu(C_9_H_8_N)_3_P (2), are shown in Figure 4e. After grinding, the intermolecular π···π interactions between adjacent ligands were breached, leading to a 42.2% and 13.4% decrease in QY of complexes 1 and 2, respectively [37]. Recently, Bakr et al. have observed a crystallization-induced emission enhancement phenomenon in [Cu_15_(PPh_3_)_6_(PET)_13_]^2+^ (PET = 2-phenylethanthiol, PPh_3_ = triphenylphosphine) [38]. When Cu NCs are dissolved, they present almost non-luminescence; however, crystallization makes the molecules luminescent (Figure 4f). The QY value of the crystalline state is 3.2%, which is much higher than that of the solution state (0.1%). It is worth pointing out that the “tri-blade fan” structure configuration plays an important effect in controlling the structural rigidity. In addition, the compact C-H···π and π···π interactions of ligands jointly contributed to the AIE performance.
Figure 4(**a**) Graphic illustration of self-assembly based on coordination effect of Zn^2+^ ions with different binding levels. Reprinted/adapted with permission from Ref. [32]. 2017, copyright Royal Society of Chemistry. (**b**) Photoemission spectra of ATT-AuNCs in different pH buffers. (**c**) Photoluminescence decay of ATT-AuNCs and Sc^3+^-ATT-AuNCs. (**d**) The binding model of Sc^3+^ to ATT-AuNCs. Reprinted/adapted with permission from Ref. [33]. 2022, American Chemical Society. (**e**) PL spectra of crystalline and ground samples of complexes 1 and 2. Reprinted/adapted with permission from Ref. [37]. 2019, copyright Royal Society of Chemistry. (**f**) PL spectra of [Cu_15_(PPh_3_)_6_(PET)_13_]^2+^ NCs in crystalline and solution state. Reprinted/adapted with permission from Ref. [38]. 2021, copyright Wiley-VCH.
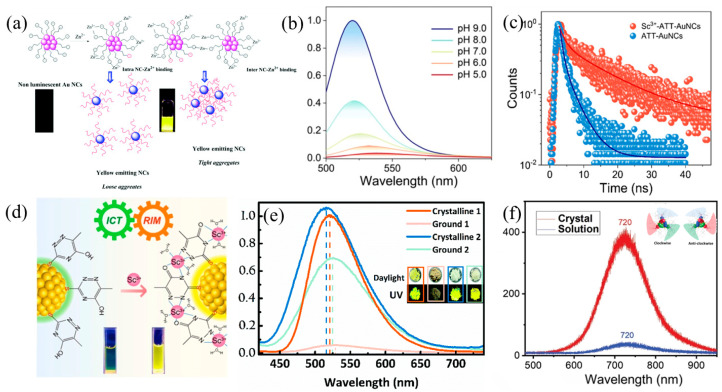



### 2.4. pH-Induced AIE

The pKa of ionizable groups strongly affects the structure and activity of molecules, which determines the state of the complex in water: solution, gel, or suspension. In other words, the aggregation of metal NCs would be pH-dependent in most cases. Bimetallic Au-Cu GSH NCs (GSH: glutathione reduced), for example, show a typical pH-dependent AIE phenomenon [39]. They are weakly fluorescent in neutral or alkaline EtOH/aqueous solution, whereas regulating the pH < 5 produces intense radiation at 584 nm. This aggregation is assigned to the COO^−^ and NH^3+^ interactions between electron-donating groups in a lower pH. However, in neutral or alkaline conditions, negative charges exhibit strong electrostatic repulsion, resulting in the dispersion of clusters. As a result of structural similarity, the pH-induced AIE phenomenon of GS/C-Au NCs (GS/C: glutathione and citrate) was discovered by Tao’s group, which benefits from the changes in surface charge density [40]. In those NCs, three noticeable features were observed in the process of regulating pH: (i) the luminescence intensity of GS/C-Au NCs was positively correlated with their surface charge density at 4.1 < pH < 5.6; (ii) when pH > 8.66, the luminescence intensity began to decrease due to the formation of carboxylate groups; (iii) when pH > 9.12, the Au(I)···Au(I) interaction was destroyed, because the bonds between the thiol group and Au(I) begin to dissociate, resulting in a further decrease in luminescence. Huang et al. explored the AIE mechanism by adjusting the pH to determine the charge state of ligands on Cu NCs [41]. As the pH was adjusted to alkaline, the structure of Cu NCs changed from the aggregate to the solution state, and the luminescence performance decreased significantly. The high pH makes the ligand of Cu NCs form a deprotonated state (COO−), which suppresses the formation of aggregates. Similarly, Ag NCs have been verified to achieve aggregation-induced emission based on the same mechanism. A typical case was exemplified by Zhou et al. in 2018, who demonstrated that the weak emission in TSA-Ag NCs (TSA: thiosalicylic acid) was dramatically improved in an acidic environment [42]. In summary, the pH-induced AIE offers a simple and efficient route to optimize the performance of metal NCs.

### 2.5. AIE Inheritance from Ligands

Using the AIE properties of organic ligands to endow clusters with AIE activity is an ordinary and remarkable strategy to construct AIE-type metal NCs. 3,5-dimethyl4-(4-(1,2,2-triphenylvinyl)benzyl)-1H-pyrazole (DTBP), a typical AIE luminogen, was selected as an organic ligand for the synthesis of the polymeric clusters M_3_(DTBP)_3_ (M = Cu, Ag or Au) by Mak and co-workers [43]. These clusters showed similar blue-green emission and lifetimes and a strong AIE response in solution. In the solid state, a 7.55-fold QY boosting was found (from 1.59% to 12.01%). From the perspective of structure, each pyrazolyl group connects two metal ions by µ_2_–η_1_, η_1_ ligation type, which constructs a stable triangular structure. Moreover, molecular layers are stacked in an ordered arrangement, with an interlayer spacing of Cu_3_, Ag_3,_ and Au_3_ of 3.37, 3.26, and 3.45 Å, respectively (Figure 5a). A stable structure restricts the intermolecular motions, thus reducing the non-radiative decay and improving AIE intensity. 4-(3,5-dimethyl-1H-pyrazol-4-yl) benzaldehyde (HL) was selected as a ligand to form clusters, because its structure is similar to DTBP [44]. Cu^I^-HL exhibits AIE behaviors with a 20-fold fluorescent enhancement when it aggregates, which is ascribed to the multiple intermolecular hydrogen bonds and rigid structure. Very recently, Perruchas et al. prepared two [Cu_4_I_4_L_4_] copper iodide clusters, with PPh_2_(C_6_H_4_-CH_2_OH) and PPh_2_(C_3_H_7_) as ligands, showing AIE photophysical behaviors in solution and obviously enhanced luminescence intensity in the solid state [45]. Interestingly, the decay lifetime was significantly prolonged, and the biexponential was transformed to monoexponential with the formation of a solid-state phase, which can be ascribed to the suppression of non-radiative relaxation in a rigid state.

### 2.6. Surface Constraint-Induced AIE

As a key component of AIE-type metal NCs, the restraining action of surface ligands can give a more rigidified molecular conformation, thereby preventing energy loss and maximizing emission efficiency. Thus, the better AIE performance of clusters could be triggered by adding mutually constrained ligands. A representative example is Ag_29_(BDT)_12_(TPP)_4_ NC (BDT: 1,3-benzenedithiol; TPP: triphenylphosphine), reported by Zhu et al. in 2018, which exhibited obvious red fluorescence [46]. However, adding additional TPP in the DMF of Ag_29_(BDT)_12_(TPP)_4_ can significantly increase the luminous intensity (about 13-fold, from 0.9% to 11.7%) (Figure 5b). The main reason for this phenomenon is that the TPP dissociation–aggregation stage on the cluster surface is prevented, due to the excess TPP ligands. The as-paired ligands/molecules rigidify the out-shell ligands and motifs to reduce nonradiative relaxation decay in LMCT and LMMCT processes. Based on the strong intermolecular and ion-pairing forces between glutathione (GSH) and T-tetraoctylammonium (TOA), the Lee group achieved rigidity of the Au_22_(SG)_18_ shell (Figure 5c) [10]. The measured QY value for the Au_22_-TOA NCs was 62%, which is nine-fold higher than that of Au_22_ NCs (~7%). The better fluorescence behavior in Au_22_-TOA is attributed to the suppressed nonradiative relaxations by TOA cations, which is demonstrated by the extended decay lifetime (from 380 ns to 2.44 μs). In 2017, the Chen group incorporated L-arginine (Arg) on 6-aza-2-thiothymine (ATT)-protected Au NCs to yield supramolecular host–guest assemblies [47]. As shown in Figure 5d, a strong hydrogen-bonding interaction was formed based on the guanidine group of Arg and ATT, keeping these clusters’ surface rigid and suppressing the intramolecular rotation and vibration of ATT. As a result, the Arg/ATT-Au NCs show an excellent QY (65%), which is much better than that (1.8%) of ATT-Au NCs.

Endowing ligands with multiple interaction sites to anchor metal cores can effectively enhance the structural rigidity of metal NCs. Zhang et al. obtained two Ag(I) carbonyl clusters: PMVEM-Ag NCs and PMAA-Ag NCs (PMVEM: polymethyl vinyl ether-alt-maleic acid; PMAA: poly(methacrylic acid)), which showed distinct AIE photophysical behaviors [48]. Both PMVEM-Ag NCs and PMAA-Ag NCs show a short decay lifetime (1.0 ns) and poor QY values (1.0%) in water solution. The precise control of the aggregation degree is realized by adding dimethyl sulfoxide (DMSO) as a solvent. For PMVEM-Ag NCs, as DMSO is gradually added to water, the spectral peak gradually splits from a broad emission band (500 nm) to two peaks (460 and 530 nm). Moreover, the intensities of these two new peaks are enhanced by 3-fold and 54-fold, respectively. In particular, the 530 nm emission peak shows a long decay lifetime (97.1 μs) and higher QY (40%), which is different from the intrinsic fluorescence. Hydrogen bond evolution is the main factor affecting the aggregation of carbonyl groups. As shown in Figure 5e, in water, the interaction of carbonyl with a metal core is significantly weaker than that of water and carboxyl, which inhibits the aggregation of carbonyl. Meanwhile, in the DMSO solution, the strong hydrogen bond interactions are weakened, creating dense aggregation of the carbonyl groups on the surface of the metal nucleus, finally resulting in a strong electron delocalization effect. For PMAA-Ag NCs, DMSO fails to achieve delocalized electrons’ conjugation between carbonyl groups because of the large steric hindrance effect of methyl groups in PMAA.

### 2.7. Mineral-Confined AIE

Minerals, with unique topology, uniform micro-pores, and abundant acid–base sites, can effectively anchor metal NCs’ surface ligands to improve the AIE. Zeolites are an excellent candidate, because their spatial structure can be easily adjusted by changing coordination properties, total charge, and surface counter-ions. Inducing silver NCs into the rigid cavities of zeolites to trigger excellent luminescence properties has been an effective strategy. As an example, (Ag_0.5_Na_6_)^+^[Al_6.5_Si_17.5_O_48_] (referred to as FAUY[Ag_0.5_]) was prepared by calcining the combined parent zeolite and silver nitrate aqueous solution at 450 °C in the air [49]. The PLQY rises to near unity based on the regulation of non-framework metal cations, which is the highest QY of Ag clusters reported so far in the zeolite framework. Ag_4_(H_2_O)_x_^2+^ was presented by Marta and co-workers in 2018; its structure is shown in Figure 6a [50]. This structure produces long decay lifetime triplet emission from a superatom quantum system constructed by hybridized silver atoms and oxygen orbitals. Lievens et al. studied this material in detail, including the crystal structure and the ultrafast electron dynamics [51]. Temperature-induced emission tuning of silver-loaded zeolite from green to white was reported by the Li group in 2019 [52]. It was clear that the luminescence color was related to the Ag state: green emission comes from Ag_3_*^n^*^+^, and white emission comes from (Ag_2_)^+^ and (Ag^+^)_2_.

Layered double hydroxides (LDHs) with adjustable interlayer space can anchor guest molecules (such as metal NCs) in periodic long-range ordered arrays. The confinement of LDH can promote aggregation and restrict the movement of clusters, further enhancing the electron–hole pairs’ recombination efficiency and fluorophore luminescence intensity. The first report of 2D LDH-confined gold NCs dates back to 2015, when (Au NCs/LDH)*_n_* films were obtained by a layer-by-layer assembly process (Figure 6b) [53]. Much stronger QY (from 2.69 to 14.11%) and a longer average decay lifetime (from 1.84 to 14.67 μs) were observed for (Au NCs/LDH)*_n_* complexes. In detail, the COO^−^ group in Au NCs was adsorbed by LDH nanosheets due to hydrogen bonds and host–guest electrostatic interactions, which inhibit the rotation and vibration of Au NCs, further resulting in boosted optical performance. In the following year, the same group constrained Au NCs using 2D exfoliated layered double hydroxide (ELDH) nanosheets to attain a QY of 19.05% [54]. Chromotropic acid (CTA) and LDH-confined Ag NCs (CTA-AgNCs/LDH) are another representative case of mineral-confined AIE of metal NCs, reported by Jin et al. in 2016 [55]. Based on the confinement effect of host–guest interactions, the CTA-AgNCs/LDH shows a high QY of 12.08% with an emission peak at 565 nm, which is much better than CTA-Ag NCs with a QY of 2.15% at 540 nm. Moreover, the Jia group observed a 14-fold QY improvement (from 0.11 to 14.27%) after achieving the assembly of Cu NCs and LDH [56]. The improved fluorescence intensity was attributed to the electron–hole being trapped in the quantum well structure of LDH.

### 2.8. MOF-Confined AIE

Metal-organic frameworks (MOFs) have proven to be an excellent carrier for metal NCs due to their ideal pore size, high loading capacity, abundant surface ligands, and cluster-based metal nodes. Metal NCs were rigidified in MOFs, which effectively limited nonradiative transition caused by intramolecular vibration and rotation, finally generating intense luminescence. A classical and interesting case presented by Zang et al. demonstrated the anchoring effects of MOF well [57]. By using 4,4′-bipyridine (bpy) to replace the CH_3_CN ligands, [(Ag_12_(S*^t^*Bu)_8_(CF_3_COO)_4_(bpy)_4_)]*_n_* (Ag_12_bpy) with high stability (>1 year) and QY (12.1%) was constructed. From the perspective of structure, the bidentate ligands of bpy act as bridges to link Ag_12_ clusters to form a double-layer structure in the *a* and *b* directions. Moreover, the double-layer structure is stacked in the c direction, forming an ordered arrangement and a structurally rigidified composite framework (Figure 6c). The enhanced luminescence was achieved mainly through the following factors: (i) inhibited nonradiative relaxation decay; (ii) improved metal Au^I^ to ligand bpy charge transfer (MLCT), and (iii) improved ligand (S, O) to ligand (bpy) charge transfer (LLCT). Further, Zang introduced a −NH_2_ group based on Ag_12_bpy to obtain a fluorescence-phosphorescence dual-emission cluster [58]. The 456 nm blue emission originates from the amino group, and the yellow emission at 556 nm comes from the phenyl groups. The lone-pair electrons of −NH_2_ strengthen spin–orbit coupling, resulting in the phosphorescence lifetime increasing from 0.2 μs to 3.12 ms. In 2019, the Tang group reported an Ag cluster (1⊃DMAC, DMAC = dimethylacetamide) from the self-assembly of the AIEgens 1,1,2,2-tetrakis(4-(pyridin-4-yl)phenyl)-ethene (tppe) and silver chalcogenolate cluster [59]. Interestingly, 1⊃DMAC showed a fluorescence color transition from blue (470 nm) to green (532 nm) when exposed to the atmosphere. Moreover, the fluorescence could return to blue immediately with treatment with DMAC. The free rotation of conjugated groups (e.g., phenyl and pyridyl) being restricted in the MOF is the fundamental reason for this reversible fluorescence change.

Among various MOFs, Zeolitic Imidazolate Framework-8 (ZIF-8) is considered a promising candidate for encapsulating metal NCs. Two representative examples are Au_25_(SG)_18_@ZIF-8 and Au_25_(SG)_18_/ZIF-8, reported by the Shi group in 2018 [60]. The Au_25_(SG)_18_@ZIF-8 was prepared by intercalating Au_25_(SG)_18_ into the ZIF-8, and Au_25_(SG)_18_/ZIF-8 was achieved by impregnating Au_25_(SG)_18_ NCs on the surface of ZIF-8 (Figure 6d). For Au_25_(SG)_18_@ZIF-8, the orientations of thiolate ligands were strictly restrained, thus showing similar luminescence as Au_25_(SG)_18_ solids. Concretely, it exhibited red emission centering at 680 nm and a weak infrared emission. For Au_25_(SG)_18_/ZIF-8, its fluorescence resembled Au_25_(SG)_18_ NCs’ solution, with a broad near-infrared emission (700–800 nm) accompanying a 665 nm red emissive. This behavior was attributed to the coordination effect between Zn^2+^ ions and surface carboxyl groups. Other representative examples of ZIF-8-based NCs have also been reported, such as Ag NCs-BSA@ZIF-8 [61], Ag NCs/ZIF-8 [62], Au NCs/ZIF-8 [63], etc.
Figure 6(**a**) Diagram of a hydrated Ag_4_ NC-contained sodalite zeolite cage. Reprinted/adapted with permission from Ref. [50]. 2018, copyright American Association for the Advancement of Science. (**b**) Schematic representation of AuNCs’ luminescence enhancement. Reprinted/adapted with permission from Ref. [53]. 2015, copyright Wiley-VCH. (**c**) Mechanism model of the preparation of Ag_12_bpy NC-based metal-organic framework. Reprinted/adapted with permission from Ref. [57]. 2017, copyright Springer Nature. (**d**) Mechanism model of the synthesis processes for Au_25_(SG)_18_@ZIF-8 and Au_25_(SG)_18_/ZIF-8 nanocomposites. Reprinted/adapted with permission from Ref. [60]. 2017, copyright Wiley-VCH.
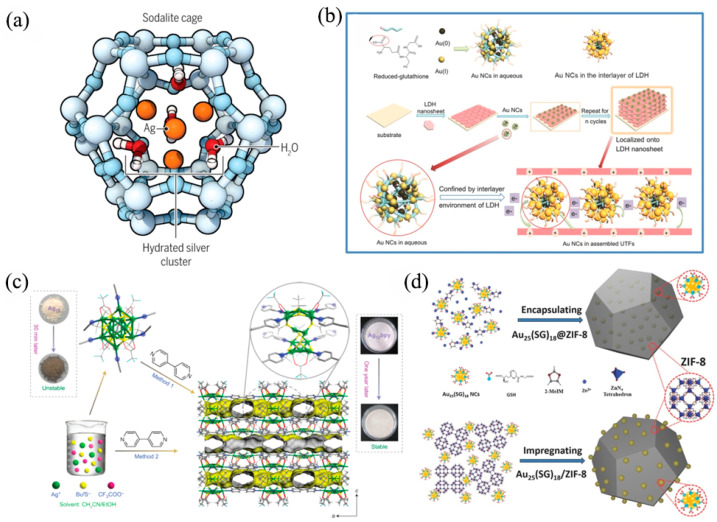



## 3. Applications of AIE-type metal NCs

AIE-type metal NCs retain both the excellent properties of metal NCs and AIE activity, attracting a large amount of research on cutting-edge applications. Over the past decade, plentiful sensory systems and optoelectronic devices have been studied based on AIE-type metal NCs, including bioimaging, photodynamic therapy, light-emitting diodes (LEDs), detection assays, circularly polarized luminescence (CPL), etc. In this section, we will briefly describe the recent developments in this field.

### 3.1. Bioimaging

The high emission efficiency of AIE-type metal NCs (especially Au NCs), combined with superior biocompatibility and little cytotoxicity, make them ideal candidates for bioimaging. Among them, peptides and alkaloid-capped Au NCs are promising [64,65,66,67,68]. For example, the Cheng group showed the potential in deep tissue visualization by employing NIR-II Au_25_(SG)_18_ (SG: glutathione) as translatable probes [69]. The Au_25_(SG)_18_ significantly combines with hydroxyapatite (HA) of the bone matrix and shows the bone structure in vivo in high resolution and contrast. Due to the ultrasmall size, these Au NCs could be rapidly excreted and hardly accumulate in the liver and spleen. Qu and coworkers designed nucleic acid-driven AIE-type Au NCs for telomerase detection with favorable sensitivity [70]. As shown in Figure 7a, strand A was modified on the surface of Au NCs through sulfhydryl groups and hybridized with telomerase substrate oligonucleotide (TS primer). In addition, the hairpin structure will be opened and connected to strand B-modified Au NCs under the impact of telomerase, resulting in the aggregation of Au NCs and thus enhanced fluorescence. In this vein, significantly, the in situ visualization of telomerase activity in cells and in vivo can be achieved.

Based on pH sensitivity, Au NCs can be used to sense and monitor intracellular pH [71]. Cao et al. confined Au NCs in MOFs to hold aggregation and limit the rotation of ligands to enhance luminescence. Moreover, this complex was applied for real-time monitoring of drug release in vivo due to its pH-dependent luminescence (Figure 7b) [63]. Au NCs-MOF is stable in neutral and alkaline environments but quickly decomposes in acidic environments, resulting in reduced luminescence. Au NCs-MOF can transport drugs (e.g., CPT: camptothecin) into cancer cells. Drugs and Au NCs escaped from the MOF due to acidic environments (pH = 5–6) when CPT@Au NCs-MOF was close to cancer cells. Thus, drug release can be monitored in real time via Au NCs luminescence. Ligand-functionalized Au NCs can be used to target intracellular high reactive oxygen species (hROS, e.g., •OH, ClO^−^, and ONOO^−^) to prevent oxidative stress diseases [72]. Au NCs protected by quaternary ammonium with oligopeptides as a linker have been developed by Jiang et al. to be utilized for intracellular imaging (Figure 7c) [73]. The AuNCs became smaller, and their valence changed from Au(0) to Au(I) after the hROS effect, which caused the loss of fluorescence. To improve sensitivity and selectivity, the Quan group developed dual-emission ratiometric fluorescent probes, CNC@GNCs RFP, consisting of Au NCs, cellulose nanocrystals (CNCs), and nonluminous o-phenylenediamine (o-PD) [74]. Au NCs emit blue fluorescence, and o-PD can be oxidized through hROS to form 2, 3-diaminophenazine (o-PDox) to emit yellow fluorescence. Thus, based on the action of hROS, this probe can achieve an obvious color transition. Zebrafish experiments confirm the probe’s potential for biological applications in both tissue penetrability and ROS responsiveness (Figure 7d–f).
Figure 7(**a**) Mechanism model of telomerase-induced nanocluster assembly. Reprinted/adapted with permission from Ref. [70]. 2021, copyright Royal Society of Chemistry. (**b**) Mechanism model of drugs encapsulated into Au NCs-MOF. Reprinted/adapted with permission from Ref. [63]. 2017, copyright Royal Society of Chemistry. (**c**) Mechanism model of the one-step preparation of red-emitting Au NCs. Reprinted/adapted with permission from Ref. [73]. 2018, copyright Wiley-VCH. (**d**–**f**) Selectivity of CNC@GNCs RFP for hROS in vivo of different Zebrafish. Reprinted/adapted with permission from Ref. [74]. 2022, copyright Elsevier.
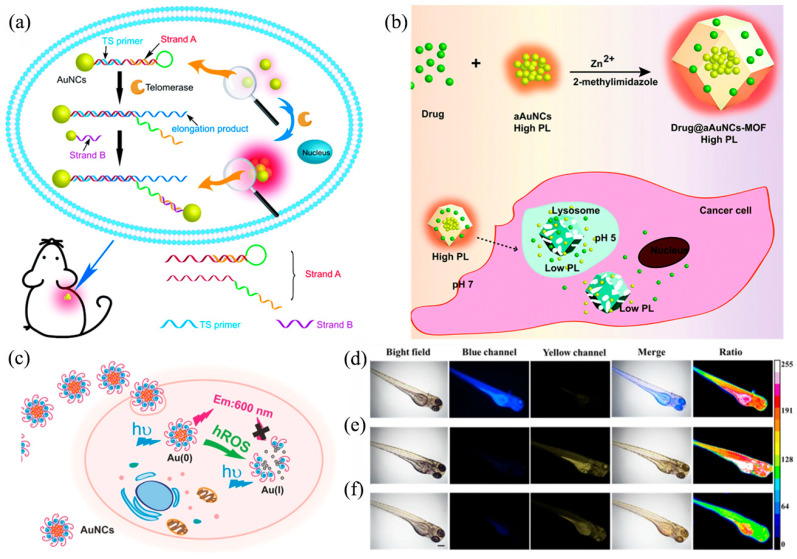



### 3.2. Biological Diagnosis and Therapy

Besides bioimaging, AIE-type metal NCs are promising candidates for biological diagnosis and therapy, including antibacterial agents, cancer radiotherapy, and so on. 

As early as 2013, the Xie group demonstrated that glutathione (GSH)-protected Ag NCs can generate a high concentration of reactive oxygen against the bacteria pseudomonas aeruginosa [75]. Furthermore, a higher antibacterial activity of Ag(I) than Ag(0) was revealed, and the corresponding Ag NCs have been applied against gram bacteria [76]. In 2016, antibiotic-grafting Ag NC technology was developed, raising the level of antibacterial ability (Figure 8a) [77]. D-Ag NCs can effectively destroy the bacterial membrane to form larger pores and continuously generate ROS, resulting in strong damage to bacterial DNA. There are also other antibacterial agents, such as Au NCs, which could interact with bacteria to cause an imbalance in bacterial cell metabolism, resulting in increased intracellular reactive oxygen species, and thereby killing bacteria [78,79,80]. Further research found that the more negatively charged Au NCs, the more ROS produced, allowing for a better bacterial killing efficiency (Figure 8b) [81]. 

The study of AIE-type metal NCs for the diagnosis and treatment of cancer is of great value for both basic research and practical applications [82,83]. GSH-Au_29–43_(SG)_27–37_ [12], GSH-Au_25_ [84], and GSH-Au_10−12_(SG)_10−12_ [85], for example, could increase tumor uptake and targeting specificity via an improved EPR effect. Au_8_(C_21_H_27_O_2_)_8_ was developed by Zang et al. for a radiosensitizer **(**Figure 8c) [86]. Au_8_(C_21_H_27_O_2_)_8_ produced ROS after X-ray exposure, causing irreversible cell apoptosis. Through rational structural design, AIE-type metal NCs can be used for photodynamic therapy (PDT) to achieve more effective anticancer treatments. Liu and coworkers have designed amine-terminated, PAMAM dendrimer-encapsulated Au NCs, which can consume H_2_O_2_ through catalase in the physiological pH range [87]. The possible mechanism was attributed to the fact that tertiary amines are easily protonated in acidic solutions, resulting in pre-adsorption of OH on the metal surface, thereby promoting catalase-like reactions. NIR-II-triggered PDT shows significantly increased tissue penetration, bypassing the limitations of conventional PDT. In the alkyl-thiolate Au NC@HSA/CAT system (HAS: serum albumin and CAT: catalase), the Au NCs can generate singlet oxygen to trigger PDT under 1064 nm laser excitation. Further, the HSA can improve the physiological stability of the nanoparticles. CAT is also used to enhance the efficacy of PDT by triggering the decomposition of tumor endogenous H_2_O_2_ into oxygen (Figure 8d–e) [88]. In addition, other biomedical applications of Au NCs have also been explored (e.g., for Parkinson’s disease treatment). N-isobutyryl-L-cysteine (L-NIBC)-protected Au NCs significantly reversed dopaminergic (DA) neuron loss in substantia nigra and striatum, preventing the fibrillation of α-Synuclein (α-Syn) [89].
Figure 8(**a**) Schematic illustration of D-Ag NCs damaging bacteria. Reprinted/adapted with permission from Ref. [77]. 2016, copyright American Chemical Society. (**b**) Influence model of surface properties of Au NCs on antibacterial activity. Reprinted/adapted with permission from Ref. [81]. 2018, copyright American Chemical Society. (**c**) Au_8_NCs for cancer radiotherapy via the ROS burst. Reprinted/adapted with permission from Ref. [86]. 2019, copyright American Chemical Society. (**d**) Mechanism model of prepare Au NC@HSA/CAT nanoparticles. (**e**) In vivo fluorescence images of tumor-bearing nude mice taken at different time points. Reprinted/adapted with permission from Ref. [88]. 2018, copyright Springer Nature.
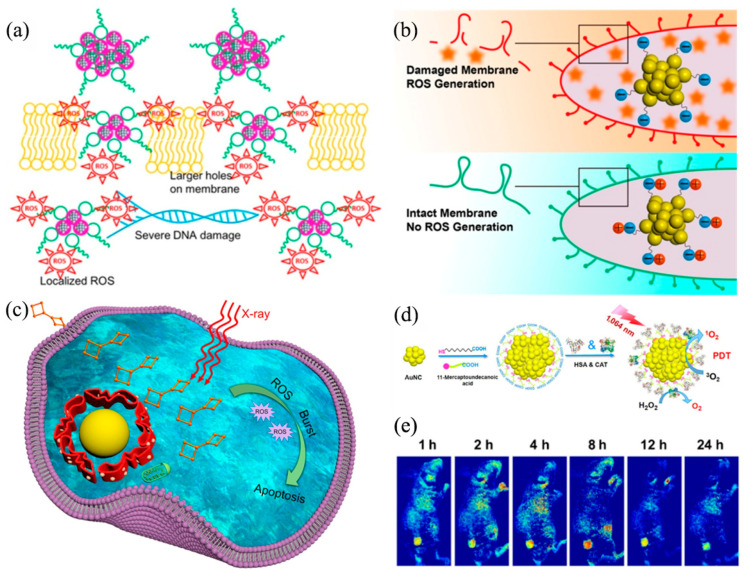



### 3.3. Light-Emitting Diodes (LEDs)

With the advantages of high luminescence efficiency, excellent durability, and low production costs, LEDs have led to innovations in the entire lighting industry. Normally, white light can be achieved by combining UV/n-UV/blue chips with polychromatic phosphors [90,91,92,93]. The AIE-type metal NCs are highly emissive in the solid state and are thus promising for LED phosphor applications [50,52]. Through ordered structure self-assembly technology, AIE NCs with emission colors can be easily designed and generated. A white device with color coordinates at (0.32, 0.36) was built by coating Cu and Au NCs on a 365 nm GaN LED chip [94]. Zn-coordinated glutathione-stabilized NCs (ZnGsH-Au NCs) have been reported for conversion layers. The device performance of its assembled LED showed (0.38, 0.38) of the CIE index and a CRI value of 75 [94]. CPL NCs can also be prepared for LED devices. For example, a circularly polarized light source was successfully realized based on copper(I)-iodide hybrid clusters [95]. Highly quantum-efficient (>95%) chiral silver clusters Ag_6_L_6_ were prepared into a WLED by combining 465 nm blue LEDs, which emitted nearly identical white emission as commercial lighting systems (blue chips + YAG: Ce^3+^) [15].

Besides pc-WLED, the Zang group demonstrated that Au NCs (e.g., (Au_4_L_4_)_n_/(Au_4_D_4_)_n_) also have significant application potential in OLEDs [96]. (Au_4_L_4_)_n_/(Au_4_D_4_)_n_) was fabricated into multilayer OLED devices with the configuration poly(3,4-ethylenedioxythiophene): poly(styrenesulfonic acid) (PEDOT: PSS) (50 nm)/enantiomers (50 nm)/1,3,5-tris(N-phenylbenzimidazol-2-yl)benzene (TPBi) (50 nm)/8-hydroxyquinolinolato-lithium (LiQ) (2 nm)/Al by a solution process. The OLEDs had a maximum EQE and dissymmetry factor of 1.5% and 1.12 × 10^−2^, respectively. Koen Kenne’s group reported ZEOLEDs based on silver-exchanged zeolites [97]. They showed that the key factors affecting the device performance were the concentration of metal NCs and the degree of zeolite anode contact. Last year, Li and coworkers introduced aggregation-enhanced Cu_3_(L_2_)_3_ (L_2_ = 4-hexyl-3,5-dimethyl-1Hpyrazole) acting as an emissive layer into OLEDs [29], where the OLED exhibited red emission centering at 610 nm, with a current efficiency (of 7.21 cd/A), a luminance (1200 cd/m^2^), as well as EQE (4%).

### 3.4. Detection Assays

Because of the sensitivity of luminescence to the surrounding environment, AIE-type metal NCs can be used as fluorescent probes to detect metal ions, small organic molecules, polypeptides, amino acids, and proteins. For example, Au^+^ ions have a high affinity for Hg^2+^ ions, which leads to the luminescence quenching of NCs [98]. Based on this principle, a series of clusters were prepared, such as lysozyme- [99], cytidine- [100], and protein-Au NCs [101] for the detection of Hg^2+^ ions, and the range of the detection limit was 0.5–60 nM. Similarly, metal NCs can detect other metal ions such as Cu^2+^, Pb^2+^, Ag^+^, Fe^3+,^ and so on [102,103,104,105]. 

The luminescence of Au NCs heavily relies on the valence state of Au, thus enabling the detection of small molecules with redox properties. Chang et al. used cholesterol oxidase (ChOx) to reduce cholesterol to H_2_O_2_ and then quenched the luminescence of BSA-Au NCs, thereby allowing for the quantitative detection of cholesterol. This approach can detect cholesterol ranging from 1 to 100 μM, with a detection limit of 1.4 mΜ [106]. Glucose and doxycycline have also been detected through analogical means [107,108]. By impregnating the Au NCs into glycol-chitosan (GC) nanogel, Au NCs@GC was developed as an H_2_S detector by Wang et al. in 2018 [109]. The confinement effect significantly improved sensitivity to aqueous sulfides. For amino acid and protein detection, metal NCs are suitable due to the interaction between the clusters and the detected object. The surface defects of Au NCs could be modified by cysteine to enhance luminescence, which allows for the quantitative detection of cysteine [89]. Based on the static quenching and inner filter effect (IFE), the fluorescent “on-off-on” of CQDs/Au NCs was achieved for detecting Cd^2+^ and L-ascorbic acid [110]. Protein enzymes can be detected through protein-Au NCs, because the luminescence signal intensity of Au NCs will change when enzymes degrade the protein templates [111,112].

### 3.5. Circularly Polarized Luminescence (CPL)

Polarization, as an intrinsic property of light, is a prerequisite for three-dimensional optical display, remote sensing, spin information optical communication, circular polarization tomography, and information encryption. Typically, CPL can be obtained when non-polarized light passes through a quarter-wave plate. However, energy loss and complex device structure restrict the popularization of this method. As an alternative efficient pathway, chiral luminescent materials were used to produce CPL directly [113]. The ideal CPL materials should have both a high quantum efficiency (QE) and a large *g_lum_* value. The *g_lum_* value means the luminescence dissymmetry factor, which can be defined by Equation (1) [114,115]:(1)$$g_{lum}=\frac{2(I_{L}-I_{R})}{I_{L}+I_{R}}$$
where *I_L_* and *I_R_* represent the luminescence intensity of left- and right-CPL, respectively. A high QE indicates efficient energy conversion, whereas the performance of common organic compound-based emitters is always poor due to the ACQ effect [116,117]. Fortunately, the advent of AIE materials opens a window for CPL studies [118,119]. With the researchers’ efforts, AIE-type metal NCs have become a member of the CPL materials family.

Recently, Yao et al. worked on a design for highly efficient chiral hybrid copper(I) halides NCs [95,120,121]. As shown in Figure 9a, R/S-Cu_2_I_2_(BINAP)_2_ hybrid clusters with a layered conformation were designed and synthesized [121]. Benefiting from the biphosphine chelating coordination of chiral ligands, the stability of the chiral cluster was greatly improved and could be dissolved in polar solvents such as dimethyl sulfoxide (DMSO) to achieve solution processing. Further, with the assistance of polyvinylpyrrolidone (PVP), hexagonal flake-like microcrystals with a high *g_lum_* (9.5 × 10^−3^) were obtained in an ethanol solution based on intermolecular interactions (Figure 9b). On the basis above, Yao et al. synthesized high-performance CPL materials (PLQY: 32%, *g_lum_*: 1.5 × 10^−2^) with the biomimetic non-classical crystallization (BNCC) strategy [95]. The main crystallization process was revealed, involving nanocrystal nucleation, aggregation, oriented attachment, and mesoscopic transformation (Figure 9c). Moreover, electrostatic interactions and Van der Waals forces can induce these aggregated nanoparticles to assembly and chiral adjustment. Thus, chiral polycrystals with different morphologies can be achieved by adjusting the chiral center position, chain length, and concentration of ligands. In addition, (R/S-MBA)_2_CuCl_2_ [122], (1-M/1-P)_2_Cu_2_I_2_ [123], (R/S-MBA)_4_Cu_4_I_4_ [124], and other chiral copper(I) halides also enrich the database of CPL materials.

Zang et al. incorporated (R/S)2,2-bis(diphenylphosphino)-1,1-binaphthyl on a BuSCu copper source to yield the CPL material R/S-Cu_3_ [125]. In the aggregated state, R/S-Cu_3_ displays notable circularly polarized luminescence (*g_lum_*: 2 × 10^−2^) due to the chiral structure and AIE feature. The luminescence comes from metal cluster-centered (MCC) and triplet metal-to-ligand charge-transfer (^3^MLCT) processes and is limited by the intramolecular motion mechanism. Following that, in 2020, the authors further developed a copper cluster of [Cu_14_(R/S-DPM)_8_](PF_6_)_6_ (R/S-Cu_14_) by modifying (R/S)-2-diphenyl-2-hydroxylmethylpyrrolidine-1-propyne ligands [126]. The R/S-Cu_14_ dilute solution was non-luminescent and CPL inactive. However, by choosing n-hexane as the solvent, the copper clusters formed well-dispersed colloids of dense aggregates, showing strong red emission and CPL signals (*g_lum_*: 1 × 10^−2^), as shown in Figure 9d. This luminescence enhancement is attributed to electrostatic interaction, C−H···π, C−H···F, and O−H···F interactions of ligands, which could substantially restrict the intramolecular rotations and vibrations. Besides Cu NCs, ultrasmall Au NCs, (e.g., L/D-[Au_4_(C_9_H_8_S_2_N)_4_], abbreviated as Au_4_PL_4_ and Au_4_PD_4_; and L/D-{[Au_4_(C_6_H_10_S_2_N)_4_]_3_}_n_, abbreviated as (Au_4_L_4_)_n_ and (Au_4_D_4_)_n,_ also have significant CPL characteristics [96]. Both Au_4_PL_4_/Au_4_PD_4_ and (Au_4_L_4_)_n_/(Au_4_D_4_)_n_ exhibited bright luminescence and were CPL active (*g_lum_* = 7 × 10^−3^) after adding poor solvent (H_2_O), with the QE increasing by 6.5% and 4.7%, respectively (Figure 9e).
Figure 9(**a**) Mechanism model of the single crystal growth process of R/S-Cu_2_I_2_(BINAP)_2_ hybrid materials (left) and image of the R-Cu_2_I_2_(BINAP)_2_ single crystal (right). (**b**) SEM image of R-Cu_2_I_2_(BINAP)_2_ hexagon platelet-shaped microcrystals. Reprinted/adapted with permission from Ref. [121]. 2021, copyright American Chemical Society. (**c**) Mechanism model of the biomimetic non-classical crystallization process. Reprinted/adapted with permission from Ref. [95]. 2022, copyright Springer Nature. (**d**) Images of R-Cu_14_ in different n-hexane solutions. Reprinted/adapted with permission from Ref. [126]. 2020, copyright Wiley-VCH. (**e**) Images of (Au_4_L_4_)_n_ in dimethylformamide with 0–90% of H_2_O. Reprinted/adapted with permission from Ref. [96]. 2020, copyright Springer Nature.
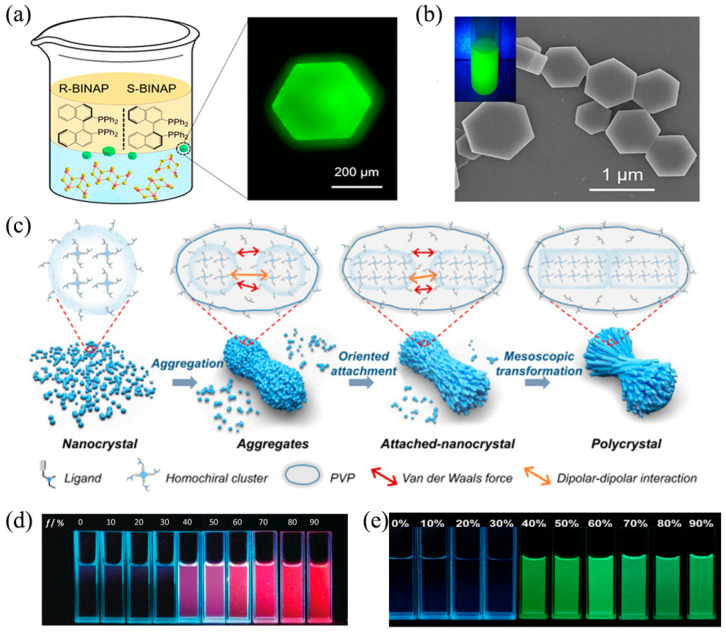



## 4. Conclusions and Outlook

Looking back, AIE metal NCs have taken the first step in terms of preparation techniques and practical applications. However, the current development of cluster chemistry has not yet reached an unambiguous agreement on the AIE fundamentals of metal NC luminescence. Moving forward, the research on AIE-type metal NCs will remain a scientific hotspot. Herein, we elaborated our perspectives, including but not limited to:-Composition: Except for a few structural units for the synthesis of metal NCs, most organic ligands and noble metals (e.g., Au, Ag, and their alloys) are scarce and/or expensive. To accommodate large-scale production for commercial purposes, the development of low-cost metal NC materials is necessary. For instance, the AIE properties of copper, zinc, and other transition metal NCs are also worth designing for a breakthrough, because of their similar electronic configurations (d^10^s^1^) and wide range of oxidation states.-Structure: Regarding the development of metal NCs’ structural isomers, the following issues need to be addressed urgently in the future: (i) how to synthesize and stabilize novel structural isomers; (ii) how to rationally separate and structurally identify isomers. For this, DFT calculation, SXRD, ESI-MS, UV-vis-NIR absorption spectroscopy, and thin-layer chromatography techniques would help to address this issue; (iii) accurately mapping the internal relationship between structure and AIE properties at the molecular level, which will lay the foundation for prediction, clipping, and preparation of excellent AIE metal NCs.-Property: The QY directly affects the future of metal NCs for application and commercialization. In particular, some red- and infrared-emitting metal NCs still have a large gap with commercial phosphors, which limits their application. Moreover, metal NCs easily decompose or lose their luminescent activity, exhibiting terrible thermo-stability and chemical stability under thermal and environmental stimuli. Some strategies, such as conferring strong chemical bond forces between the metal core and ligands, using MOF protection, etc., have improved stability. These measures are efficient to a certain extent but cannot solve all the problems of this issue.-Mechanism: The current development of cluster chemistry has not yet reached an unambiguous agreement on the AIE fundamentals of metal NC luminescence, such as: (i) the correlation between metal NCs’ structural characteristics and AIE behaviors; (ii) AIE concepts at the branch level in metal NCs such as aggregation-induced emission enhancement (AIEE) and clusterization-triggered emission (CTE); (iii) a clear identification of similar overlapping topics between AIE-type metal NCs and other metal counterparts. For example, the aggregation of metal nanoparticles supports the emission of adjacently located emitter moieties, which has been well-developed in systems such as cryosoret and soret colloid nano-assembly [127,128,129,130].-Applications: To explore the application potential of AIE metal NCs in related biological and medical systems, the development of water-soluble AIE metal NCs with near-infrared emission or long afterglow luminescence is still in its infancy. In addition, there is an urgent need to develop metal NC-based optoelectronics devices to serve practical applications in various sectors. Last but not least, the preparation of high-quantum yield AIE metal NCs is necessary to develop their applications in the latest technologies (such as photonic crystal-coupled emission for biosensors).

The AIE mechanism has been widely accepted and continuously improved upon in the research community of luminescent metal NCs. However, quoting Laroche, to “have achieved unremittingly, indomitable than when subjected to failure is more important”. AIE and/or metal NCs researchers still have a long way to go to witness the fundamentals and applications of AIE-type metal NCs. We are hopeful that this review will accelerate the advancement of such promising research, as well as provide models or springboards for many pending challenges in other cross-disciplines and research fields.

## Figures and Tables

**Figure 1 nanomaterials-13-00470-f001:**
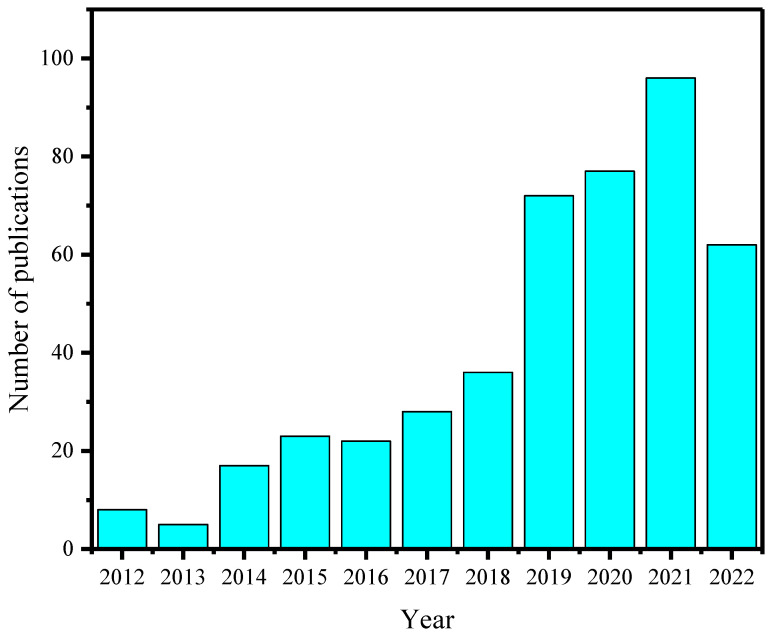
The increasing number of publications on AIE-type metal NCs over the years between 2012 and 2022 (data statistics from the Web of Science in January 2023, using keywords of AIE and metal nanoclusters).

**Figure 2 nanomaterials-13-00470-f002:**
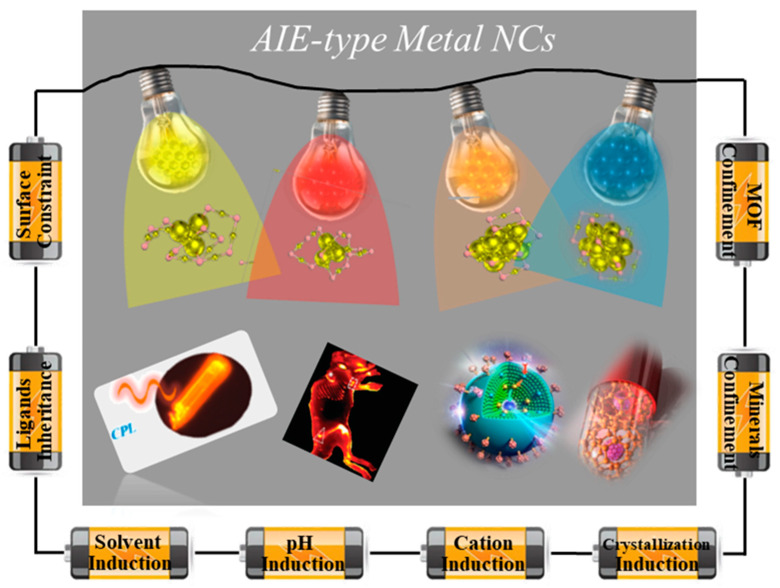
Schematic illustration of AIE-type metal NCs from the design strategies to the applications.

**Figure 3 nanomaterials-13-00470-f003:**
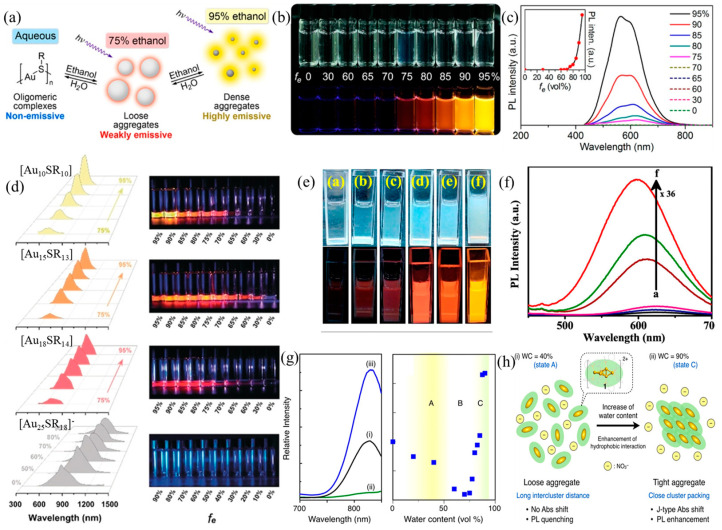
(**a**) Mechanism model of solvent-induced AIE phenomenon of Au(0)@Au(I)-SR NCs. (**b**) Digital photos and (**c**) photoemission spectra of Au(0)@Au(I)-SR with different *f_e_*. Reprinted/adapted with permission from Ref. [17]. 2012, copyright American Chemical Society. (**d**) Luminescence spectra (**left**) and the corresponding digital photos (**right**) of Au_10_SR_10_, Au_15_SR_13_, Au_18_SR_14,_ and [Au_25_SR_18_]^−^ with a different fraction of ethanol and water. Reprinted/adapted with permission from Ref. [26]. 2020, copyright Wiley-VCH. (**e**) Digital photos and (**f**) PL spectra of Cu_34−32_(SG)_16−13_ with different *f_e_*. Reprinted/adapted with permission from Ref. [27]. 2019, copyright American Chemical Society. (**g**) Photoluminescence spectra of (i) MeOH, (ii) MeOH/water (30/70), and (iii) MeOH/water (10/90). (**h**) Mechanism model of possible aggregate forms of MeOH/water at low (i) and high (ii) WC. Reprinted/adapted with permission from Ref. [28]. 2020, copyright American Chemical Society.

**Figure 5 nanomaterials-13-00470-f005:**
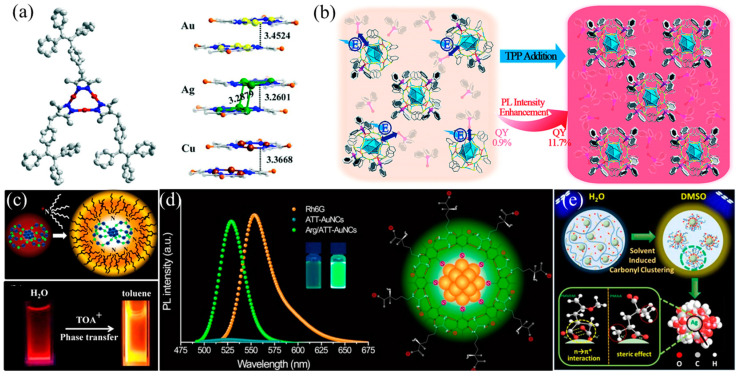
(**a**) Schematic representing the crystal structure of M_3_ and interlayer stacking structures of Au_3_, Ag_3,_ and Cu_3_. Reprinted/adapted with permission from Ref. [43]. 2019, copyright Royal Society of Chemistry. (**b**) Mechanism model of TPP dissociation-aggregation process. Reprinted/adapted with permission from Ref. [46]. 2018, copyright Royal Society of Chemistry. (**c**) Mechanism model of binding TOA to Au_22_(SG)_18_ and a digital photograph of Au_22_(SG)_18_ and TOA-Au_22_. Reprinted/adapted with permission from Ref. [10]. 2015, copyright American Chemical Society. (**d**) Mechanism model of the Arg-mediated ATT-AuNCs. Reprinted/adapted with permission from Ref. [47]. 2017, copyright American Chemical Society. (**e**) Mechanism model of the solvent-induced stage of Ag NCs. Reprinted/adapted with permission from Ref. [48]. 2017, copyright American Chemical Society.

## Data Availability

The study did not report any data.

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
