# Peer review of "Viewing Aggregation-Induced Emission of Metal Nanoclusters from Design Strategies to Applications"

_nanomaterials, 2023, doi:10.3390/nano13030470_

Round 1

Reviewer 1 Report

Please find attached the reviewer comments attached as a separate PDF.

Reviewer 2 Report

The review with the title “Viewing Aggregation-Induced Emission of Metal Nanoclusters  from Design Strategies to Applications” by Tingting Li, Haifeng Zhu, and Zhennan Wu leaves me in an ambivalent state.

On the one side, I really like the topic. It covers a most interesting and growing field of material science. The coverage seem quite complete and I have no real problems to agree with the authors on their appraisal and outlook.

On the other hand, I have to conclude that the paper suffers from a lack of a decent introduction into the physical foundations of the AIE phenomenon. Instead, the fact that the suppression of rotational and vibrational processes increases the luminescence properties is mention on numerous occasions.

The main problem of the paper is nevertheless the very poor English. It seems to me that the manuscript has either been written by a person with a very limited command of the English language, or that it has been translated into English using Google translator or a similar tool. In any case, the reading process is most cumbersome as basic grammatical rules are violated in almost every sentence.
Although in many sentences the intention of the authors can be guessed, there are also many sentences, which left me absolutely clueless. It is the latter problem which is most serious, as it requires a cooperation of the authors with a person with sufficient grasp of English to transform the manuscript into something that can be published.

My impression is that the language problem is so serious that it cannot easily be resolved by the journal’s editorial staff. I therefore cannot support acceptance of the manuscript as is. It certainly requires a major revision, which should include a revised introduction and a most thorough language polishing.

Round 2

Reviewer 1 Report

Authors have addressed the reviewer comments satisfactorily.

Reviewer 2 Report

I like the addition to the introduction, expaling what the AIE phenomenon is.
The quality of the English is also improved. While I still feel that it could be better I think this can be handled easily by the editorial staff.